# Microstructural Evolution and Mechanical Properties of Graphene Oxide-Reinforced Ti6Al4V Matrix Composite Fabricated Using Spark Plasma Sintering

**DOI:** 10.3390/nano11061440

**Published:** 2021-05-29

**Authors:** Ying Song, Weiwei Liu, Yufeng Sun, Shaokang Guan, Yao Chen

**Affiliations:** 1School of Materials Science and Engineering, Zhengzhou University, Zhengzhou 450001, China; songying55555@163.com (Y.S.); yfsun@zzu.edu.cn (Y.S.); skguan@zzu.edu.cn (S.G.); 2School of Mechanical & Electric Engineering, Soochow University, Suzhou 215006, China; liuweiwei@suda.edu.cn; 3Key Laboratory of Advanced Magnesium Alloys in Henan Province, Zhengzhou 450002, China; 4Key Laboratory of Advanced Materials Processing & Mold, Ministry of Education of the People’s Republic of China, Zhengzhou 450002, China

**Keywords:** graphene oxide, nanocomposite, titanium matrix, microstructure, mechanical properties, spark plasma sintering

## Abstract

To achieve a further reduction in weight of titanium alloys and to satisfy the increasing demand of energy-saving for aerospace and automotive applications, a graphene oxide nanosheet-reinforced Ti6Al4V (GO/TC4) composite was successfully fabricated using spark plasma sintering (SPS). Contrary to the Widmanstätten microstructure of a monolithic TC4 sample, the microstructure of the composites displayed a typical basket-weave structure in virtue of the introduced residual tensile stress generated from the mismatch of coefficients of thermal expansion (CTE) between GO and TC4 during the phase transformation. Meanwhile, the in situ-formed TiC nanolayer and diffusion layer were identified at the GO–TC4 interface, which is expected to endow a stronger interfacial bonding. As compared with the TC4 sample, the TC4 composite with the addition of 0.27 wt.% GO exhibited a 0.2% yield strength of 921.8 MPa, an ultimate tensile strength of 1040.1 MPa, and an elongation of 5.3%, displaying a better balance of strength and ductility than that of the composite with a higher GO addition (0.54 wt.%). The synergetic strengthening mechanisms such as Orowan strengthening, enhanced dislocation density strengthening, and load transfer were confirmed. Among them, load transfer contributed greatly to the strength of the composites due to improved interfacial bonding between the GO fillers and TC4 matrix.

## 1. Introduction

Titanium alloys have been widely recognized as important and promising engineering materials used in diverse industrial fields spanning aerospace, weaponry, automobiles, biomedicine, and others, because they exhibit a good combination of low density [1], high specific strength [2], excellent corrosion resistance [1], and biocompatibility [3]. Among these titanium alloys with different compositions, the Ti6Al4V alloy is dominant in the aerospace market, making up about 60% of the titanium used in jet engines and up to 80–90% for airframes [4,5]. However, it should be noted that actual applications of the Ti6Al4V alloy in the aerospace and automotive applications are still facing some limitations due to its poor adhesive wear resistance associated with its low hardness and much lower strength than those of high-strength steel [6,7]. Hence, significant efforts have been devoted to develop Ti6Al4V-based composites reinforced by various ceramics such as TiC [8], TiB [9], TiC-TiB [10], SiC [11], ZrO_2_ [12], and Ti_5_Si_3_ [13], in which TiB whisker and TiC particles have been well demonstrated to be the optimal reinforcements due to the fact that they not only exhibit a high modulus and hardness, but also possess similar densities and coefficients of thermal expansion (CTE) to Ti6Al4V alloy, subsequently contributing to the reduction in residual stresses at the interface of the designed composites [14,15]. For example, Huang et al. [15] successfully fabricated tailor-designed titanium matrix composites with the reticular reinforcement architecture of TiC_p_ or TiB, imparting the composites with an enhanced strengthening effect in a wide range of temperatures.

For a further reduction in weight of titanium alloys to satisfy the increasing demand of energy-saving in aerospace and automotive applications, it is imperative to develop titanium-based composites reinforced by novel reinforcing agents with a higher strength and lower density compared to those of the conventional ceramics. Graphene is a two-dimensional nanomaterial comprising a single-layer of sp^2^-hybridized carbon atoms and exhibits excellent mechanical and physical properties, including a high tensile strength (~130 GPa), high Young’s modulus (~1.0 TPa), large surface area (~2630 m^2^/g), and very low density (~1.8 g/cm^3^) [16,17,18], making it more suitable as a reinforcing nanofiller to improve the mechanical properties of metal materials [19]. As compared to the well-known one-dimensional carbon nanotube (CNT), selecting graphene as a reinforcement has the following advantages: (1) graphene is much more cost-effective than CNT and can be produced on a large scale; (2) graphene with a higher surface area is prone to form stronger interfacial bonding with a metal matrix.

It should be noted, however, that the great challenge in graphene-reinforced metallic composites is to tailor the homogeneous distribution of these 2D nanofillers in the matrix in a suitable manner, because the strong van der Waals force between the multilayer graphene (MLG) usually forms graphene clusters, subsequently leading to stress concentration, acting as crack nuclei [7]. To this end, the current powder mixing methods applied in the graphene-reinforced metal composites are usually ball milling [20,21] and/or ultrasonic dispersion [22,23] because of their simplicity, time demands, and cost efficiency. Nevertheless, a high impact force and shear force along with ball milling, especially with high-energy ball milling (HEBM), have been reported to reduce the graphene size and to destroy its structural integrity, which is believed to easily introduce defects and impurities in the original graphene structure [24,25]. As such, the improvement in mechanical properties of graphene-reinforced metal matrix composites usually fails to achieve the level of graphene/polymer composites.

As an important derivative of graphene, graphene oxide (GO) with reactive groups populating its edge (carbonyl and carboxyl groups) and planes (hydroxyl and epoxide groups) has been demonstrated to possess desirable dispersion behavior in aqueous media [26]. Moreover, GO possesses good mechanical properties [27] and is easier to produce on a large scale than graphene is. Such typical characteristics associated with GO hold great promise for serving as nanofillers in the development of high-performance metallic composites. Although some investigations have been conducted on the GO-reinforced pure titanium composites, little information focusing on the efficacy of the GO on the microstructure evolution and mechanical properties of Ti6Al4V composites is available. Herein, the aim of the present work, as an extension of our previous investigation [28], was to reveal the influence of GO on the matrix microstructure, interfacial structure, and mechanical properties of Ti6Al4V-based composites fabricated using spark plasma sintering (SPS), which is beneficial for their practical industrial applications.

## 2. Materials and Methods

Argon atomized pre-alloyed Ti6Al4V (hereafter referred to as TC4) powders (Falcon Technology Co., Ltd., Wuxi, China) with an average size of ~15–60 μm were used as precursor material in this research. The morphology of the TC4 powders was observed by scanning electron microscopy (SEM, Hitachi S-4700, Tokyo, Japan), as depicted in Figure 1a. GO agglomerated powders (purity: 99.9%) with a thickness of ~0.55–1.2 nm (Chengdu Organic Chemicals Co. Ltd., Chengdu, China) were used as the reinforcing nanofillers, as shown in Figure 1b.

The electrostatic self-assembly method was employed to homogeneously disperse GO in the TC4 powders, in which the GO dispersion solution with a negative charge is expected to uniformly adhere to the surface of the TC4 powder with a positive charge, functionalized through the addition of cetyltrimethyl ammonium bromide (CTAB). GO agglomerates (0.65 g) were added into deionized water (250 mL) for 30 min of ultrasonication, which was subjected to zeta-potential measurement (Zetasizer Nano ZS90 DLS, Malvern, UK). The zeta potential of the GO solution was −21.6 mV. It should be noted that the GO powders were all heated to 120 °C under vacuum conditions prior to the preparation of GO suspension. Figure 1c shows the GO nanosheet after ultrasonication, and it was found that as-received GO agglomerates were effectively exfoliated to several layers, and its rough and wrinkled surface was also noticeable. On the other hand, CTAB was added into deionized water to obtain a 0.1 wt.% CTAB aqueous solution, and it was stirred at 50 °C until the solution became transparent. Afterward, TC4 powders (130 g) were added into the CTAB aqueous solution (130 mL) followed by ultrasonication for 10 min. The CTAB-TC4 slurry was centrifuged by refrigerated centrifuge (Allegra X-15R, Beckman Coulter, Pasadena, CA, USA) for 20 min at 3000 r/min, and it was then washed twice with deionized water. The zeta potential of the CTAB-functionalized TC4 suspension was measured to be 38.6 mV. Finally, the CTAB-TC4 suspension and GO dispersion were fully mixed through magnetic stirring. Stationary standing of the mixed slurry was employed until the color became transparent. The mixed slurry was filtered and rinsed, followed by vacuum drying at 100 °C for 8 h to obtain the composite powders. Figure 1d shows an SEM micrograph of the as-prepared powder mixture where the platelet-like GO adhered to the surface of these TC4 spherical particles in virtue of the electrostatic self-assembly process. For comparison, TC4, 0.5 wt.% GO-TC4, and 1.0 wt.% GO-TC4 composite powders were employed in this research.

A spark plasma sintering (SPS) system (LABOX-330, SINTER LAND, Niagata, Japan) was used to fabricate monolithic TC4 samples with and without GO addition. The GO-TC4 powder mixture was compacted at 1100 °C under a pressure of 30 MPa in a graphite die (inner diameter: 30 mm, height: 40 mm). A heating rate of 50 °C/min was used for raising the fabrication temperature up to 1100 °C, followed by an isothermal holding time of 30 min. The samples were then cooled down to ambient temperature with a cooling rate of the furnace. It should be noted that graphite papers were placed between the powders and die/punches for easy specimen removal. For comparison, the monolithic TC4 sample was fabricated following the same processing.

Phase identification of the sintered samples was carried out through X-ray diffraction (XRD, X’Pert-ProMRD, Almelo, Holland) with Cu Kα radiation using a scanning rate of 20°/min at a range of 20 to 90°. Metallographic samples were prepared using standard procedures and etched with a solution consisting of 7 mL of H_2_O, 2 mL of HF, and 1 mL of HNO_3_. Microstructure observations of the sintered samples were carefully examined using optical microscopy (OM, 4XB, Shanghai Taiming Optical Instrument Co., Ltd., Shanghai, China), SEM, and high-resolution transmission electron microscopy (HRTEM, Tecnai 20 G2, FEI The Netherlands). The relative densities of SPS-fabricated samples were determined based on the Archimedes principles, and each sample was measured at least five times to achieve repeatability. The thermal stability of GO was measured using a thermal gravimetric analyzer (TGA, Pyris-1, PE, USA) in the temperature range of 25–800 °C with a heating rate of 10 °C. Fourier-transform infrared (FT-IR) of the as-received GO and SPSed composites was performed using a VERTEX 70+HYPERION 2000 apparatus (Bruker, Karsruhe, Germany).

Tensile specimens with a cross-section of 2.0 × 1.0 mm and a gauge length of 15 mm were machined by wire electric discharge machining from the as-sintered sample. Tensile tests were conducted on each sample under a constant cross-head speed with an initial strain rate of 1 × 10^−3^/s using an Instron 5566 testing machine.

## 3. Results and Discussion

### 3.1. Microstructure Characteraziation of the Composites

The thermal stability of as-received GO measured using TGA is shown in Figure 2a, and it demonstrates that GO first experienced a slight mass reduction (~12%) when it was heated up to ~120 °C, which should be closely related to water loss. In the second stage (120 °C–250 °C), there was a significant mass loss (~28%) of GO, corresponding to decomposition of the oxygen-containing groups (hydroxyl and carboxyl) [25]. Finally, a further mass loss (~18%) of GO occurred due to the fact that epoxy groups along with GO started to decompose at 270 °C. As stated before, the as-received GO powders were preserved at 120 °C under vacuum conditions prior to preparation of the GO suspension. Therefore, it is reasonable that the actual content of the added GO retained in the SPSed composites was only ~54% as compared to that in the as-prepared composite powders, i.e., the weight fractions of the addition of GO in the composites were 0.27 wt.% for 0.5 wt.%GO/TC4 composite powders and 0.54 wt.% for 1.0 wt.%GO/TC4 composite powders (hereafter represented as 0.27GO/TC4 and 0.54GO/TC4, respectively).

FT-IR spectra of the as-received GO and SPSed composite are depicted in Figure 2b. As for the as-received GO powders, the peak observed at a wavenumber of 1049 cm^−1^ is related to the stretching vibration of the C-O-C band, and the signal at 1213 cm^−1^ is assigned to the stretching vibration of the C-O bond in epoxy groups. The peak that appeared at 1729 cm^−1^ should correspond to the C=O band in the carboxyl and carbonyl groups, and the broad peak between 3100 and 3700 cm^−1^ is due to the stretching vibration of the O-H bond from hydroxyl groups and/or water molecules. Additionally, the peak at 1619 cm^−1^ ascribes to the stretching vibration of the C=C band. In contrast, the adsorptions of C-H, O-H, and C-O bonds that appeared in the FT-IR spectrum of the sintered GO/TC4 composite were significantly weaker than those of as-received GO powders. Hence, the TGA and FI-TR results sufficiently demonstrate that the addition of GO was partially reduced to rGO in the SPS process.

As shown in Figure 3a, it is clear that the sintered TC4 sample displayed a typical Widmanstätten microstructure, in which several colonies consisting of alternate layers of the α phase and thin secondary β phase were randomly oriented in the primary β phase. It is also interesting to observe that the primary β grains grew rapidly in such a way that their size was significantly larger than that of the as-received TC4 powders (15–60 μm), which is likely due to the fact that several as-received TC4 powders were consolidated through high pressure along with the SPS process and new primary β phase developed from the merged β phase. In striking contrast, the microstructure of the sintered composites developed into α laths and an intergranular β phase, in which the α laths intersected each other to form a typical basket-weave structure (Figure 3b,c). Hence, the comparison of the microstructure leaves no doubt that the addition of GO nanosheets is capable of influencing the matrix microstructure of the sintered composites.

As for the α + β two-phase titanium alloys, it is well known that the Widmanstätten microstructure is significantly controlled by elastic strain energy generated from the different specific volumes of the new phase and parent phase during phase transformation [29], i.e., lower elastic strain energy often allows for a Widmanstätten microstructure in the process of slow cooling from above the β transus temperature (~990 °C) associated with SPS parameters employed in this research. In the composites, it should be pointed out that GO exhibits a much smaller value (−8.0 × 10^−6^ K^−1^) [27] of the coefficient of thermal expansion (CTE) than that of TC4 alloy (9.5 × 10^−6^ K^−1^) [30], which would produce a high residual tensile stress within the Ti matrix in the vicinity of these nanofillers. Thus, the residual tensile stress is believed to generate a higher elastic strain energy of the matrix, subsequently warding off the formation of a Widmansätten microstructure in the composite.

XRD patterns of SPSed samples are depicted in Figure 4a, and it is clear that the addition of GO had no effect on the phase constituents of the sintered samples, i.e., all the sintered samples consisted of an α phase and β phase. Figure 4b shows the Raman spectra of the as-received GO powders and sintered composites. As compared with the main features (D and G peaks) presented in the Raman spectra, it is visible that the value of I_D_/I_G_ decreased from ~1.02 (GO powders) to ~0.82 (composites). Due to the fact that the attachment of hydroxyl and epoxide groups on the carbon basal plane usually causes the prominent D peak [31], the remarkable decrease in the values of I_D_/I_G_ associated with the sintered composites is believed to provide more evidence that functional groups of the GO powders fully or partially released during spark plasma sintering.

The interfacial structure between the reinforcement and the matrix plays a remarkable role in the mechanical properties of metal matrix composites [19]. Figure 5a shows the typical interfacial structure of the GO/TC4 composite. The presence of both GO and α-Ti was confirmed by the HRTEM images in Figure 5b,c, which is represented by the white square regions labeled b1 and b2, respectively. The measured lattice spacings of GO and α-Ti were ~0.34 and ~0.215 nm, corresponding to the (0002) crystal plane of graphene and (002) crystal plane of α-Ti, respectively. As noted in Figure 5d, the HRTEM image of the white square region labeled as b3 in Figure 5a, the localized interfacial product of the TiC nanolayer, revealed by the lattice spacing of 0.23 nm of the (200) crystal plane of TiC, was confirmed to form adjacent to the open edge of the GO nanosheet, sufficiently illustrating that open edges of GO are more reactive than its basal planes are and facilitate preferential nucleation and growth of TiC. In addition to the local interfacial reaction zone, it is evident from Figure 5e, the HRTEM image of zone b4 in Figure 5a, that a disordered layer with a width of ~1–2 nm was visible, and that this layer blurred the interface between GO and α-Ti, strongly implying that mutual diffusion of carbon and titanium atoms might be responsible for this layer.

### 3.2. Mechanical Properties of the Composites

The measured densities of the sintered samples were 4.362 ± 0.010, 4.356 ± 0.012, and 4.343 ± 0.012 g/cm^3^ for the pure TC4 sample, 0.27GO/TC4 sample, and 0.54GO/TC4 sample, respectively. On the basis of the theoretical densities of TC4 (~4.40 g/cm^3^) [32] and GO (~2.26 g/cm^3^) [27], it is obvious that the theoretical densities of 0.27GO/TiC4 and 0.54GO/TC4 composites were 4.394 and 4.389 g/cm^3^, respectively. Hence, the relative densities of the TC4 sample, and 0.27GO/TC4 and 0.54GO/TC4 composites were ~99.1%, ~99.1%, and ~99.0%, respectively. The imperceptible differences in the relative density of the sintered samples imply that these added GO nanosheets distributed homogeneously in the TC4 matrix.

Figure 6 shows the tensile stress–strain curves of SPSed TC4 samples, and the tensile properties of these samples interpreted from the engineering stress–engineering strain curves are listed in Table 1. As shown, the TC4 sample exhibited an elastic modulus of ~110.5 MPa, a 0.2% yield strength (YS) of ~790.7 MPa, and an ultimate tensile strength (UTS) of ~855.7 MPa. The GO strengthening effects varied with its addition content, which was more significant at an addition content of 0.54 wt.%, exhibiting an elastic modulus of ~133.3 MPa, yield strength of ~1044.2 MPa, and ultimate tensile strength (UTS) of ~1053.4 MPa, which were enhanced by ~22%, ~32%, and ~23%, respectively, compared with those of the monolithic TC4 sample composite. Nevertheless, it is necessary to point out that the improved strength associated with the addition of GO in the composites was achieved by compromising their elongation to failure, especially for the 0.54GO/TC4 sample, sharply decreasing to ~1.8% with respect to the TC4 sample (~6.8%). As compared to the results reported in previous investigations, the 0.25 wt.% graphene/TC4 composite fabricated using SPS at high pressures (250 MPa) displayed a YS of ~964 MPa and UTS of ~999 MPa, but its elongation to failure was only ~0.9% [7]; Dong et al. reported that the SPSed 0.15 wt.% GO/TC4 composite possessed a lower YS (~897 MPa) and UTS (~951 MPa) with a similar elongation to failure (~5.4%) [33]. Hence, it is reasonable that, in the case of graphene and/or graphene oxide-reinforced TC4 composites fabricated using SPS without post-treatments such as hot rolling and hot extrusion, the 0.27GO/TC4 composite in this research (YS of ~921.8 MPa, UTS of ~1040.1 MPa, and elongation to failure of ~5.3%, as listed in Table 1) exhibited the combination of higher YS and UTS without much sacrifice of ductility, achieving a better balance between strength and ductility.

As indicated in Table 2, it is believed that the addition of GO exerted a significant strengthening role in the TC4 composites. To understand the strengthening mechanisms of GO/TC4 composites, the following aspects should be focused: (1) Orowan strengthening of the added GO, (2) enhanced dislocation density (EDD) strengthening by mismatch in coefficient of thermal expansion (CTE) between GO and TC4, (3) load transfer from TC4 matrix to GO, (4) grain refinement pinned by GO, and (5) solid solution strengthening by carbon atoms diffused into TC4 matrix. Among these strengthening mechanisms, solid solution strengthening induced by interstitial carbon atom diffusion into the TC4 matrix can be negligible due to the fact that the further addition of carbon in titanium alloys has little effect on the improved strength when the carbon concentration is beyond its limit (~0.05 wt.% for α-Ti at ambient temperature) [34]; in addition, the addition of GO led to a remarkable difference in the microstructure (Figure 3), i.e., with the formation of a basket-weave microstructure in the composite matrix other than the Widmansätten microstructure presented in the monolithic TC4 sample, it is unsuitable to make a quantitative estimation of grain refinement on the enhanced strength of the composite samples, and therefore, the grain refinement was also ignored in this research.

Load transfer has been demonstrated to be a very important strengthening mechanism in the case of carbonaceous nanomaterial-reinforced metal composites, in which the load upon the matrix should be efficiently transferred to the reinforcing nanofillers through the interfacial shear action. As such, these nanofillers with outstanding mechanical properties are expected to bear a great portion of loading. The shear-lag model is commonly employed to estimate the load transfer strengthening efficacy in the composites, and the enhancement of the yield strength of composites (Δ*σ_LT_*) can be expressed as follows [35]:
(1)$$\begin{array}{c} \mathrm{for}\;l<l_{c}\\ \Delta \sigma_{LT}=\sigma_{GO}V_{f}\frac{l}{2l_{c}}-\sigma_{m}V_{f} \end{array}$$
(2)$$\begin{array}{c} \mathrm{for}\;l>l_{c}\\ \Delta \sigma_{LT}=\sigma_{GO}V_{f}(1-\frac{l_{c}}{2l})-\sigma_{m}V_{f} \end{array}$$
where *σ_GO_* is the yield strength of GO (24.7 GPa) [36], *σ_m_* is the matrix yield strength (790.7 MPa of the monolithic TC4 sample measured in this research), and *V_f_* is the volume fraction of GO in the composite (0.52 vol.% for 0.27GO/TC4 sample and 1.04 vol.% for 0.54GO/TC4 sample, respectively). *l_c_* is the critical length defined as [37]:(3)$$l_{c}=\sigma_{GO}\frac{Al}{\tau_{m}S}$$
where *τ_m_* is the shear strength of titanium matrix (≈0.5 *σ_m_*), and A and S are the cross-sectional area and interfacial area of GO, respectively, assuming that *A = wt* and *S = 2(w + t)l* [36], in which *w*, *t,* and *l* are the width, thickness, and length of GO, respectively. In the calculation, *l* = ~3 μm, *w* = ~3 μm, and *t* = ~ 0.01 μm were employed. Hence, the calculated *l_c_* was 0.31 μm. As such, based on Equation (2), Δ*σ_LT_* can be calculated to be ~117.7 and ~235.3 MPa for 0.27GO/TC4 and 0.54GO/TC4 composites, respectively.

In the case of GO-reinforced metal composites, these added GO nanosheets can act as obstacles to hinder dislocation movement (back stress), and the improved yield strength (Δ*σ_Orowan_*) induced by the Orowan mechanism can be calculated using [37]
(4)$$\Delta \sigma_{orowan}=0.13Gb\frac{\ln(d_{p}/2b)}{d_{p}[{\left(1/2V_{f}\right)}^{1/3}-1]}$$
where *G* is the shear modulus of the matrix (24.7 GPa) [30], *b* is the Burgers vector of TC4 (0.24 nm) [30], and *d_p_* is the reinforcement particle diameter obtained by assuming a spherical model:(5)$$d_{p}=\sqrt[{3}]{\frac{6wtl}{\pi }}$$

The estimated *d_p_* (GO) was ~0.56 μm, and the calculated Δ*σ_Orowan_* for the 0.27GO/TC4 and 0.54GO/TC4 composites were ~2.7 and ~3.7 MPa, respectively.

A significant mismatch in CTE between the reinforcement and matrix usually induces an enhanced dislocation density within the matrix adjacent to the interface, subsequently leading to strengthening. The strengthening effect can be expressed as the following equations [38]:(6)$$\Delta \sigma_{EDD}=\kappa Gb\rho^{1/2}$$
in which *κ* is the geometric constant (1.25) [27], and the density of dislocation is calculated by [39]
(7)$$\rho =\frac{BV_{f}\varepsilon }{b(1-V_{f})}\frac{1}{d_{p}}$$
where *B* is 8 for the platelet, and *ε* is expressed as [39]
(8)ε=Δα×ΔT
where ∆α is the difference between the coefficients of thermal expansion of the TC4 matrix (9.5 × 10^−6^ K^−1^) [30] and rGO (−8.0 × 10^−6^ K^−1^) [27], and ∆*T* is the difference between the sintering and test temperatures (1075 K). Therefore, the calculated Δ*_σEDD_* were 15.6 and 21.9 MPa for 0.27GO/TC4 and 0.54GO/TC4 composites, respectively.

The calculated strengthening contributions of Δ*σ_orowan_*, Δ*σ_EDD_*, and Δ*σ_LT_* are summarized in Table 2. It is well known that Orowan strengthening usually exhibits little improvement in the microsized particle-reinforced metal matrix composite due to the fact that these reinforcement particles are coarse and the inter-particle spacing is large. In this research, the lateral size of these embedded GO was up to several micrometers, which was likely ascribed to a negligible Orowan-strengthening contribution (2.7–3.7 MPa) to the composite. In contrast, EDD strengthening and load transfer were of significant importance in the TC4 composites. In particular, load transfer was greatly dominant for the enhanced yield strength (118–253 MPa) of the SPSed composites. As discussed above, these defects presented on the GO edges are capable of providing active sites for the in situ formation of a local TiC nanolayer, subsequently imparting improved interfacial bonding with the GO–TC4 interface. In addition, a diffusion layer (as depicted in Figure 5e) would also lead to a coherent interface. Hence, it is reasonable that such interfacial characteristics, along with the added GO with a large contact area with the TC4 matrix, should be greatly responsible for the improved strength of the composites.

The tensile fractured surfaces of the SPSed samples were characterized using SEM, as depicted in Figure 7. As is illustrated in Figure 7a, numerous larger dimples were visible for the pure TC4 sample, indicating that it underwent a ductile fracture mode. In the case of the 0.27 GO/TC4 composite, its fracture surface was covered with more reduced dimples (Figure 7b). Furthermore, with a further increase in GO content to 0.54 wt.%, the dimple feature gradually disappeared, and brittle fracture occurred in terms of cleavage-like failure and intergranular fracture (Figure 7c). Moreover, the presence of the micropores on the fracture surface of the 0.54 GO/TC4 composite indicated that GO agglomeration would occur in the case of higher GO contents in the composite, which, in turn, slightly decreased the density of the sintered composite.

## 4. Conclusions

A graphene oxide nanosheet-reinforced Ti6Al4V (GO/TC4) composite was successfully fabricated using spark plasma sintering (SPS), in which the electrostatic self-assembly method was employed to homogeneously disperse GO in the TC4 powders. The main findings of this research can be summarized as follows:As compared with the typical Widmanstätten microstructure in the monolithic TC4 sample, the high residual tensile stress generated from the mismatch of coefficients of thermal expansion (CTE) between GO and TC4 was responsible for the presence of the typical basket-weave in the matrix microstructure of the composites. More importantly, an in situ-formed local TiC nanolayer and diffusion layer were identified at the GO–TC4 interface.In comparison with the TC4 sample, the 0.27GO/TC4 composite exhibited an improved strength (yield strength of 921.8 MPa and ultimate tensile strength of 1040.1 MPa) and a similar elongation of 5.3%, achieving a better balance between strength and ductility.The enhanced strength is closely related to the synergetic strengthening mechanisms such as Orowan strengthening, enhanced dislocation density strengthening, and load transfer, in which the improved interfacial bonding between the nanofiller and TC4 matrix due to the presence of a TiC nanolayer and diffusion layer at the interface makes load transfer greatly responsible for the enhanced strength of the composites.

## Figures and Tables

**Figure 1 nanomaterials-11-01440-f001:**
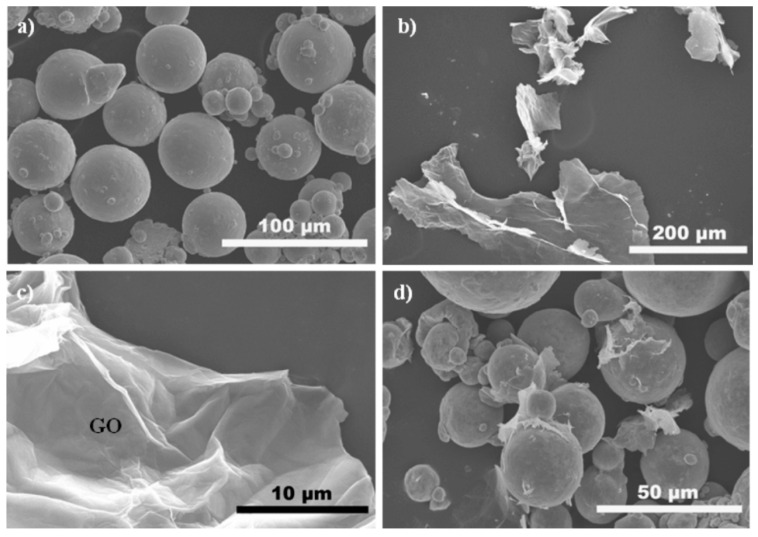
SEM images of the as-received TC4 powders (**a**), GO powders (**b**), GO nanosheet after ultrasonication (**c**), and GO-TC4 composite powders (**d**).

**Figure 2 nanomaterials-11-01440-f002:**
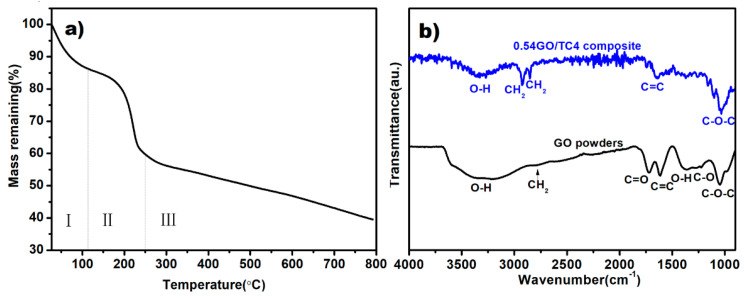
(**a**) TGA plot of the as-received GO powders, and (**b**) FT-IR spectra of as-received GO powders and as-sintered GO/TC4 composites.

**Figure 3 nanomaterials-11-01440-f003:**
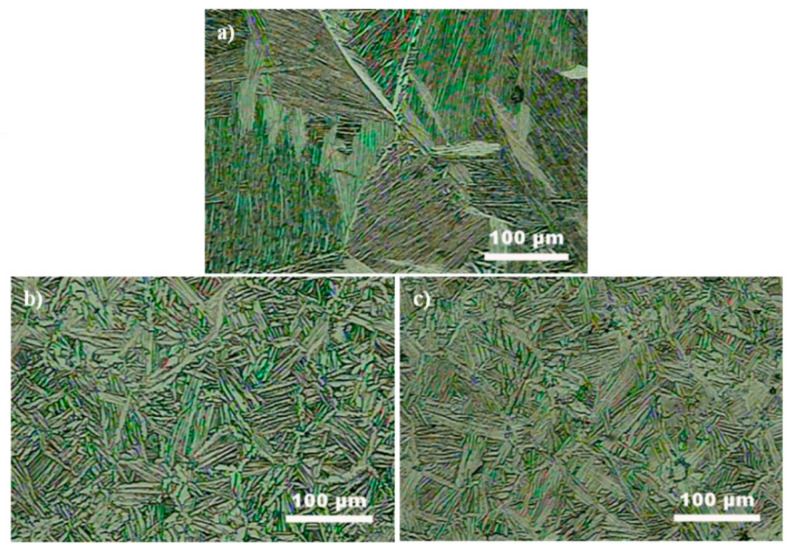
OM images showing typical microstructure of the monolithic TC4 sample (**a**), 0.27GO/TC4 composite (**b**), and 0.54GO/TC4 composite (**c**).

**Figure 4 nanomaterials-11-01440-f004:**
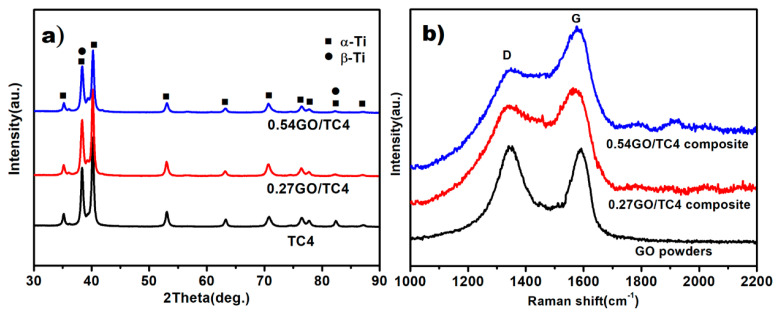
(**a**) XRD results of sintered samples, and (**b**) Raman spectra of as-received GO powders and sintered TC4 composites.

**Figure 5 nanomaterials-11-01440-f005:**
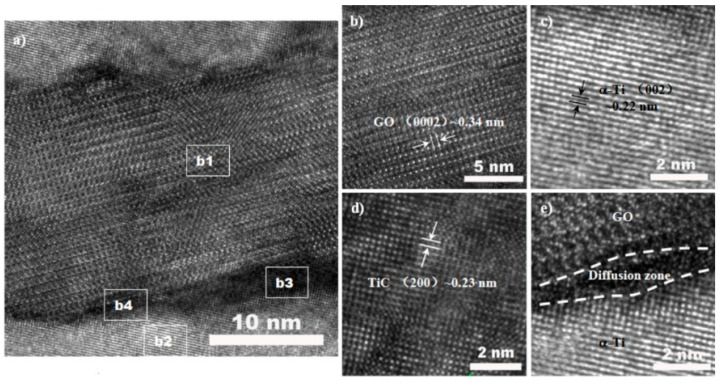
HRTEM images showing (**a**) detailed GO–TC4 interfacial structure, (**b**) the remaining GO located near the Ti matrix (the white square region b1 in (**a**)), (**c**) the corresponding lattice spacing of α-Ti and its SAED at the marked region of b2, (**d**) in situ-formed TiC nanolayer (the white square region b3 in (**a**)), and (**e**) the diffusion layer (the white square region b4 in (**a**)).

**Figure 6 nanomaterials-11-01440-f006:**
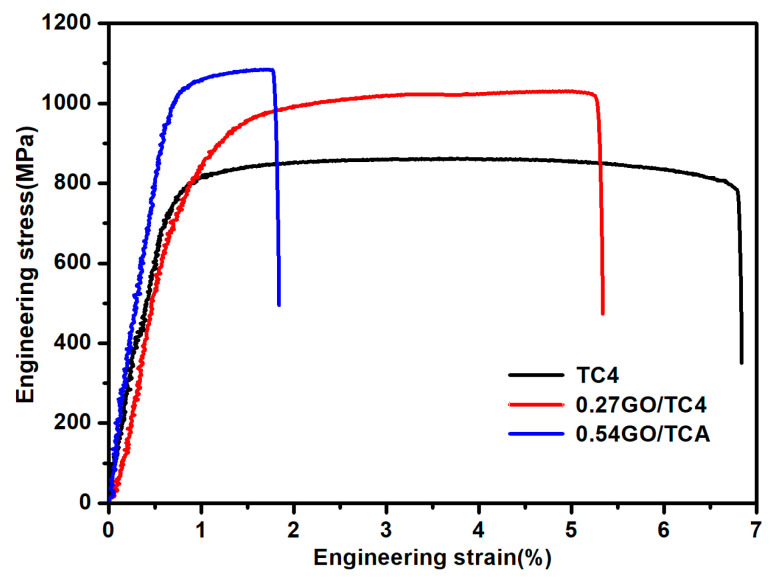
Tensile stress–strain curves of the sintered samples with different contents of GO nanosheets.

**Figure 7 nanomaterials-11-01440-f007:**
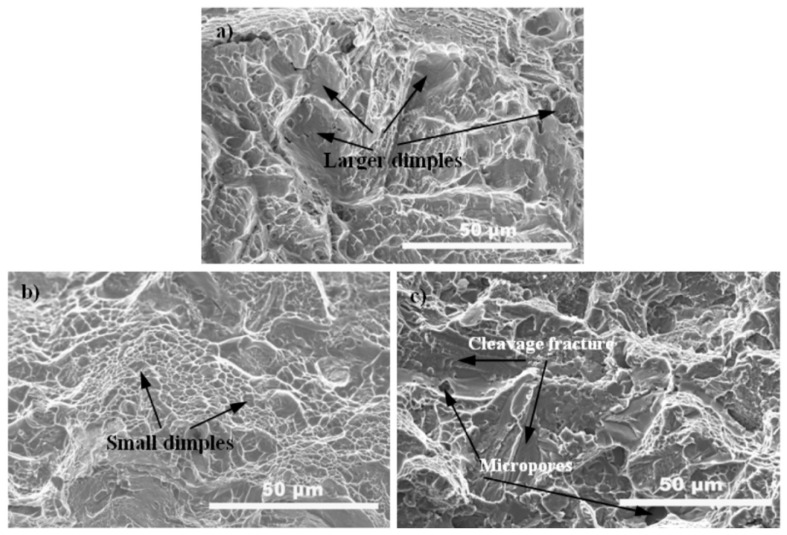
SEM images showing the tensile fracture surfaces of the monolithic TC4 sample (**a**), 0.27GO/TC4 composite (**b**), and 0.54GO/TC4 composite (**c**).

**Table 1 nanomaterials-11-01440-t001:** Mechanical properties of SPSed TC4 and GO/TC4 composites.

Samples	*E* (GPa)	*YS* (MPa)	*UTS* (MPa)	ε (%)
TC4	110.5 ± 5.4	790.7 ± 3.3	855.7 ± 15.6	6.8 ± 0.7
0.27GO/TC4	123.4 ± 3.7	921.8 ± 5.3	1040.1 ± 8.5	5.3 ± 0.6
0.54GO/TC4	133.3 ± 3.2	1034 ± 4.5	1053.4 ± 27.4	1.8 ± 0.5

**Table 2 nanomaterials-11-01440-t002:** Strengthening contributions of GO/TC4 composites (MPa).

Sample	Δ*σ_orowan_*	Δ*σ_EDD_*	Δ*σ_LT_*	Δ*σ_cal_*	Δ*σ_exp_*	Δ*σ_cal-exp_*
0.27GO/TC4	2.7	15.6	117.7	136.0	131.1	4.9
0.54GO/TiC4	3.7	21.9	235.3	260.9	243.3	17.6

where Δσ_cal_ is the total calculated strengthening contribution, and Δσ_exp_ is the increment in experimental yield strength of the composites as compared with that of the monolithic TC4 sample.

## Data Availability

The data presented in this study are available o request from the corresponding author.

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
