# Peer review of "Microstructural Evolution and Mechanical Properties of Graphene Oxide-Reinforced Ti6Al4V Matrix Composite Fabricated Using Spark Plasma Sintering"

_nanomaterials, 2021, doi:10.3390/nano11061440_

Round 1

Reviewer 1 Report

The paper is scientifically interesting and well-written. However, in order to improve its scientific quality it should be subjected to a some revision. The detailed remarks are as follows:

  1. In the following form, the Introduction could be not interesting for a reader. Please avoid blocks of references, e.g. “(...) excellent corrosion resistance and biocompatibility [1-3].”, as these do not emphasize the particular aspects from the cited papers. Particularly, when citations are made in reference to specific technical aspects, single/double, e.g. [1, 2] references are encouraged. It is strongly suggested that the references need to make in-depth comments on the content of the cited papers; avoid generic comments. Mention/comment the relevance of the cited paper and especially the research gap associated to it.
  2. Metals manufactured with the use of additive methods are very often characterized by the relatively low surface integrity and thus they require post-machining. This aspect is particularly important in a literature review, thus Authors are requested to extend the state-of-the-art by adding a new paragraph focusing on this problem. Moreover, please use - among others - the newest papers within this scope, as in example:
  • Evaluation of Surface Topography after Face Turning of CoCr Alloys Fabricated by Casting and Selective Laser Melting. Materials 2020, 13, 2448; doi:10.3390/ma13112448.
  • Geometric Specification of Non-Circular Pulleys Made with Various Additive Manufacturing Techniques. Materials 2021, 14, 1682. https://doi.org/10.3390/ma14071682
  • Surface texture formation in precision machining of direct laser deposited tungsten carbide. Advances in Manufacturing (2017) Volume 5, Issue 3, pp 251–260.

       3. In order to improve the readability of the conclusions, please formulate them in a form of a bullet points revealing the main findings both in the qualitative and quantitative way.

Author Response

Authors sincerely thank the reviewers and the editors for their constructive comments and suggestions on the manuscript entitled “Microstructural evolution and mechanical properties of graphene oxide reinforced Ti6Al4V matrix composite fabricated using spark plasma sintering” (nanomaterials-1225364) for publication in Special Issue "Mechanical and Electrical Properties of Novel Nanocomposites" in Special Issue "Nanocomposites: From Design to Application" of Nanomaterials.

We have responded to the reviewer’s comments below in an itemized manner. Also, the changes in the revised manuscript are highlighted in yellow.

Reviewer #1: The paper is scientifically interesting and well-written. However, in order to improve its scientific quality it should be subjected to a some revision. The detailed remarks are as follows:

  1. In the following form, the Introduction could be not interesting for a reader. Please avoid blocks of references, e.g. “(...) excellent corrosion resistance and biocompatibility [1-3].”, as these do not emphasize the particular aspects from the cited papers. Particularly, when citations are made in reference to specific technical aspects, single/double, e.g. [1, 2] references are encouraged. It is strongly suggested that the references need to make in-depth comments on the content of the cited papers; avoid generic comments. Mention/comment the relevance of the cited paper and especially the research gap associated to it.

Response: The authors thank the reviewer’s comment, and blocks of references have rectified in the revised manuscript.

  1. Metals manufactured with the use of additive methods are very often characterized by the relatively low surface integrity and thus they require post-machining. This aspect is particularly important in a literature review, thus Authors are requested to extend the state-of-the-art by adding a new paragraph focusing on this problem. Moreover, please use - among others - the newest papers within this scope, as in example:
  • Evaluation of Surface Topography after Face Turning of CoCr Alloys Fabricated by Casting and Selective Laser Melting. Materials 2020, 13, 2448; doi:10.3390/ma13112448.
  • Geometric Specification of Non-Circular Pulleys Made with Various Additive Manufacturing Techniques. Materials 2021, 14, 1682. https://doi.org/10.3390/ma14071682
  • Surface texture formation in precision machining of direct laser deposited tungsten carbide. Advances in Manufacturing (2017) Volume 5, Issue 3, pp 251–260.

Response: The authors thank the reviewer’s comment. As pointed out by the reviewer, metal parts fabricated using the additive manufacturing techniques usually require post-machining to improve their surface integrity. However, this manuscript focuses on the microstructure and mechanical properties of titanium composites synthesized by spark plasma sintering other than titanium composite parts fabricated using additive manufacturing techniques, and therefore the authors think that it is not necessary to add a new paragraph in the revised manuscript, focusing on the relatively low surface integrity of additively manufactured metal parts.

  1. In order to improve the readability of the conclusions, please formulate them in a form of a bullet points revealing the main findings both in the qualitative and quantitative way.

Response: The authors thank the reviewer’s comment. Main findings have been revised in a form of a bullet points in the revised manuscript.

Reviewer 2 Report

The paper deals with preparation and characterization of SPS-produced composites of Ti6Al4V and graphene oxide. Microstructure, local composition and mechanical properties are characterized and strengthening mechanisms are discussed.

Two minor technical points are marked in the pdf; so are few suggestions for minor English imperfections.

Concerning originality: There is reference 33 which is mentioned only briefly and which seems to cover exactly the same topic (only with more reinforcements). I don't have access to it at the moment to judge it in detail. It would be good to properly distinguish the current contribution and its novelty compared to ref. 33.

Author Response

Authors sincerely thank the reviewers and the editors for their constructive comments and suggestions on the manuscript entitled “Microstructural evolution and mechanical properties of graphene oxide reinforced Ti6Al4V matrix composite fabricated using spark plasma sintering” (nanomaterials-1225364) for publication in Special Issue "Mechanical and Electrical Properties of Novel Nanocomposites" in Special Issue "Nanocomposites: From Design to Application" of Nanomaterials.

We have responded to the reviewer’s comments below in an itemized manner. Also, the changes in the revised manuscript are highlighted in yellow.

Reviewer #2: The paper deals with preparation and characterization of SPS-produced composites of Ti6Al4V and graphene oxide. Microstructure, local composition and mechanical properties are characterized and strengthening mechanisms are discussed.

  1. Two minor technical points are marked in the pdf; so are few suggestions for minor English imperfections.

Response: The authors thank the reviewer’s comment. We are so sorry for the clerical errors in the manuscript. These mistakes have been rectified in the revised manuscript.

  1. Concerning originality: There is reference 33 which is mentioned only briefly and which seems to cover exactly the same topic (only with more reinforcements). I don't have access to it at the moment to judge it in detail. It would be good to properly distinguish the current contribution and its novelty compared to ref. 33.

Response: The authors thank the reviewer’s comment. Ref. 33 (Carbon 164 (2020) 272-286) reported that a Ti6Al4V (TC4) composite reinforced with 0.15wt.% graphene oxide (GO) was fabricated using an spark plasma sintering (SPS) furnace at 1173 K for 5 min of holding time under an axial pressure of 60 MPa. As compared with the SPS processing parameters employed in Ref.33, GO/TC4 composite fabricated in this research used different SPS parameters and different amount of GO addition. Therefore, the authors think that the difference in strength reported in Ref. 33 and our manuscript, respectively, might be ascribed to different microstructure of matrix in the TC4 composite. However, details on the microstructure of SPSed GO/TC4 composite were not reported in Ref. 33. Hence, it is tough to distinguish the current contribution and its novelty compared to ref. 33.